# Generalization bounds for neural ordinary differential equations and deep residual networks

**Pierre Marion**
Sorbonne Université, CNRS,
Laboratoire de Probabilités, Statistique et Modélisation, LPSM,
F-75005 Paris, France
pierre.marion@sorbonne-universite.fr

## Abstract

Neural ordinary differential equations (neural ODEs) are a popular family of continuous-depth deep learning models. In this work, we consider a large family of parameterized ODEs with continuous-in-time parameters, which include time-dependent neural ODEs. We derive a generalization bound for this class by a Lipschitz-based argument. By leveraging the analogy between neural ODEs and deep residual networks, our approach yields in particular a generalization bound for a class of deep residual networks. The bound involves the magnitude of the difference between successive weight matrices. We illustrate numerically how this quantity affects the generalization capability of neural networks.

## 1 Introduction

Neural ordinary differential equations (neural ODEs, Chen et al., 2018) are a flexible family of neural networks used in particular to model continuous-time phenomena. Along with variants such as neural stochastic differential equations (neural SDEs, Tzen and Raginsky, 2019) and neural controlled differential equations (Kidger et al., 2020), they have been used in diverse fields such as pharmokinetics (Lu et al., 2021; Qian et al., 2021), finance (Gierjatowicz et al., 2020), and transportation (Zhou et al., 2021). We refer to Massaroli et al. (2020) for a self-contained introduction to this class of models.

Despite their empirical success, the statistical properties of neural ODEs have not yet been fully investigated. What is more, neural ODEs can be thought of as the infinite-depth limit of (properly scaled) residual neural networks (He et al., 2016a), a connection made by, e.g., E (2017); Haber and Ruthotto (2017); Lu et al. (2017). Since standard measures of statistical complexity of neural networks grow with depth (see, e.g., Bartlett et al., 2019), it is unclear why infinite-depth models, including neural ODEs, should enjoy favorable generalization properties.

To better understand this phenomenon, our goal in this paper is to study the statistical properties of a class of time-dependent neural ODEs that write

$$dH_t = W_t \sigma(H_t)dt, \tag{1}$$

where $W_t \in \mathbb{R}^{d \times d}$ is a weight matrix that depends on the time index $t$, and $\sigma : \mathbb{R} \to \mathbb{R}$ is an activation function applied component-wise. Time-dependent neural ODEs were first introduced by Massaroli et al. (2020) and generalize time-independent neural ODEs

$$dH_t = W \sigma(H_t)dt, \tag{2}$$

as formulated in Chen et al. (2018), where $W \in \mathbb{R}^{d \times d}$ now denotes a weight matrix independent of $t$. There are two crucial reasons to consider time-dependent neural ODEs rather than the more

37th Conference on Neural Information Processing Systems (NeurIPS 2023).

restrictive class of time-independent neural ODEs. On the one hand, the time-dependent formulation is more flexible, leading to competitive results on image classification tasks (Queiruga et al., 2020, 2021). As a consequence, obtaining generalization guarantees for this family of models is a valuable endeavor by itself. On the other hand, time dependence is required for the correspondence with general residual neural networks to hold. More precisely, the time-dependent neural ODE (1) is the limit, when the depth $L$ goes to infinity, of the deep residual network

$$H_{k+1} = H_k + \frac{1}{L}W_{k+1}\sigma(H_k), \quad 0 \leqslant k \leqslant L-1, \tag{3}$$

where $(W_k)_{1 \leqslant k \leqslant L} \in \mathbb{R}^{d \times d}$ are weight matrices and $\sigma$ is still an activation function. We refer to Marion et al. (2022, 2023); Sander et al. (2022); Thorpe and van Gennip (2022) for statements that make precise under what conditions and in which sense this limit holds, as well as its consequences for learning. These two key reasons compel us to consider the class of time-dependent ODEs (1) for our statistical study, which in turn will inform us on the properties of the models (2) and (3).

In fact, we extend our study to the larger class of *parameterized ODEs*, which we define as the mapping from $x \in \mathbb{R}^d$ to the value at time $t = 1$ of the solution of the initial value problem

$$H_0 = x, \qquad dH_t = \sum_{i=1}^m \theta_i(t)f_i(H_t)dt, \tag{4}$$

where $H_t$ is the variable of the ODE, $\theta_i$ are functions from $[0, 1]$ into $\mathbb{R}$ that parameterize the ODE, and $f_i$ are fixed functions from $\mathbb{R}^d$ into $\mathbb{R}^d$. Time-dependent neural ODEs (1) are obtained by setting a specific entrywise form for the functions $f_i$ in (4).

Since the parameters $\theta_i$ belong to an infinite-dimensional space, in practice they need to be approximated in a finite-dimensional basis of functions. For example, the residual neural networks (3) can be seen as an approximation of the neural ODEs (1) on a piecewise-constant basis of function. But more complex choices are possible, such as B-splines (Yu et al., 2022). However, the formulation (4) is agnostic from the choice of finite-dimensional approximation. This more abstract point of view is fruitful to derive generalization bounds, for at least two reasons. First, the statistical properties of the parameterized ODEs (4) only depend on the characteristics of the functions $\theta_i$ and not on the specifics of the approximation scheme, so it is more natural and convenient to study them at the continuous level. Second, their properties can then be transferred to any specific discretization, such as the deep residual networks (3), resulting in generalization bounds for the latter.

Regarding the characteristics of the functions $\theta_i$, we make the structural assumption that they are Lipschitz-continuous and uniformly bounded. This is a natural assumption to ensure that the initial value problem (4) has a unique solution in the usual sense of the Picard-Lindelöf theorem (Arnold, 1992). Remarkably, this assumption on the parameters also enables us to obtain statistical guarantees despite the fact that we are working with an infinite-dimensional set of parameters.

**Contributions.** We provide a generalization bound for the large class of parameterized ODEs (4), which include time-dependent and time-independent neural ODEs (1) and (2). To the best of our knowledge, this is the first available bound for neural ODEs in supervised learning. By leveraging on the connection between (time-dependent) neural ODEs and deep residual networks, our approach allows us to provide a depth-independent generalization bound for the class of deep residual networks (3). The bound is precisely compared with earlier results. Our bound depends in particular on the magnitude of the difference between successive weight matrices, which is, to our knowledge, a novel way of controlling the statistical complexity of neural networks. Numerical illustration is provided to show the relationship between this quantity and the generalization ability of neural networks.

**Organization of the paper.** Section 2 presents additional related work. In Section 3, we specify our class of parameterized ODEs, before stating the generalization bound for this class and for neural ODEs as a corollary. The generalization bound for residual networks is presented in Section 4 and compared to other bounds, before some numerical illustration. Section 5 concludes the paper. The proof technique is discussed in the main paper, but the core of the proofs is relegated to the Appendix.

## 2 Related work

**Hybridizing deep learning and differential equations.** The fields of deep learning and dynamical systems have recently benefited from sustained cross-fertilization. On the one hand, a large line of work is aimed at modeling complex continuous-time phenomena by developing specialized neural architectures. This family includes neural ODEs, but also physics-informed neural networks (Raissi et al., 2019), neural operators (Li et al., 2021) and neural flows (Biloš et al., 2021). On the other hand, successful recent advances in deep learning, such as diffusion models, are theoretically supported by ideas from differential equations (Huang et al., 2021).

**Generalization for continuous-time neural networks.** Obtaining statistical guarantees for continuous-time neural networks has been the topic of a few recent works. For example, Fermanian et al. (2021) consider recurrent neural networks (RNNs), a family of neural networks handling time series, which is therefore a different setup from our work that focuses on vector-valued inputs. These authors show that a class of continuous-time RNNs can be written as input-driven ODEs, which are then proved to belong to a family of kernel methods, which entails a generalization bound. Lim et al. (2021) also show a generalization bound for ODE-like RNNs, and argue that adding stochasticity (that is, replacing ODEs with SDEs) helps with generalization. Taking another point of view, Yin et al. (2021) tackle the separate (although related) question of generalization when doing transfer learning across multiple environments. They propose a neural ODE model and provide a generalization bound in the case of a linear activation function. Closer to our setting, Hanson and Raginsky (2022) show a generalization bound for parameterized ODEs for manifold learning, which applies in particular for neural ODEs. Their proof technique bears similarities with ours, but the model and task differ from our approach. In particular, they consider stacked time-independent parameterized ODEs, while we are interested in a time-dependent formulation. Furthermore, these authors do not discuss the connection with residual networks.

**Lipschitz-based generalization bounds for deep neural networks.** From a high-level perspective, our proof technique is similar to previous works (Bartlett et al., 2017; Neyshabur et al., 2018) that show generalization bounds for deep neural networks, which scale at most polynomially with depth. More precisely, these authors show that the network satisfies some Lipschitz continuity property (either with respect to the input or to the parameters), then exploit results on the statistical complexity of Lipschitz function classes. Under stronger norm constraints, these bounds can even be made depth-independent (Golowich et al., 2018). However, their approach differs from ours insofar as we consider neural ODEs and the associated family of deep neural networks, whereas they are solely interested in finite-depth neural networks. As a consequence, their hypotheses on the class of neural networks differ from ours. Section 4 develops a more thorough comparison. Similar Lipschitz-based techniques have also been applied to obtain generalization bounds for deep equilibrium networks (Pabbaraju et al., 2021). Going beyond statistical guarantees, Béthune et al. (2022) study approximation and robustness properties of Lipschitz neural networks.

## 3 Generalization bounds for parameterized ODEs

We start by recalling the usual supervised learning setup and introduce some notation in Section 3.1, before presenting our parameterized ODE model and the associated generalization bound in Section 3.2. We then apply the bound to the specific case of time-invariant neural ODEs in Section 3.3.

### 3.1 Learning procedure

We place ourselves in a supervised learning setting. Let us introduce the notation that are used throughout the paper (up to and including Section 4.1). The input data is a sample of $n$ i.i.d. pairs $(x_i, y_i)$ with the same distribution as some generic pair $(x, y)$, where $x$ (resp. $y$) takes its values in some bounded ball $\mathcal{X} = B(0, R_{\mathcal{X}})$ (resp. $\mathcal{Y} = B(0, R_{\mathcal{Y}})$) of $\mathbb{R}^d$, for some $R_{\mathcal{X}}, R_{\mathcal{Y}} > 0$. This setting encompasses regression but also classification tasks by (one-hot) encoding labels in $\mathbb{R}^d$. Note that we assume for simplicity that the input and output have the same dimension, but our analysis easily extends to the case where they have different dimensions by adding (parameterized) projections at the beginning or at the end of our model. Given a parameterized class of models $\mathcal{F}_{\Theta} = \{F_{\theta}, \theta \in \Theta\}$, the parameter $\theta$ is fitted by empirical risk minimization using a loss function $\ell : \mathbb{R}^d \times \mathbb{R}^d \to \mathbb{R}^+$ that

we assume to be Lipschitz with respect to its first argument, with a Lipschitz constant $K_\ell > 0$. In the following, we write for the sake of concision that such a function is $K_\ell$-Lipschitz. We also assume that $\ell(x,x) = 0$ for all $x \in \mathbb{R}^d$. The theoretical and empirical risks are respectively defined, for any $\theta \in \Theta$, by

$$\mathscr{R}(\theta) = \mathbb{E}[\ell(F_\theta(x),y)] \quad \text{and} \quad \widehat{\mathscr{R}}_n(\theta) = \frac{1}{n}\sum_{i=1}^n \ell\big(F_\theta(x_i),y_i\big),$$

where the expectation $\mathbb{E}$ is evaluated with respect to the distribution of $(x,y)$. Letting $\widehat{\theta}_n$ a minimizer of the empirical risk, the generalization problem consists in providing an upper bound on the difference $\mathscr{R}(\widehat{\theta}_n) - \widehat{\mathscr{R}}_n(\widehat{\theta}_n)$.

## 3.2 Generalization bound

**Model.** We start by making more precise the parameterized ODE model introduced in Section 1. The setup presented here can easily be specialized to the case of neural ODEs, as we will see in Section 3.3. Let $f_1, \ldots, f_m : \mathbb{R}^d \to \mathbb{R}^d$ be fixed $K_f$-Lipschitz functions for some $K_f > 0$. Denote by $M$ their supremum on $\mathcal{X}$ (which is finite since these functions are continuous). The parameterized ODE $F_\theta$ is defined by the following initial value problem that maps some $x \in \mathbb{R}^d$ to $F_\theta(x) \in \mathbb{R}^d$:

$$H_0 = x$$
$$dH_t = \sum_{i=1}^m \theta_i(t) f_i(H_t) dt \tag{5}$$
$$F_\theta(x) = H_1,$$

where the parameter $\theta = (\theta_1, \ldots, \theta_m)$ is a function from $[0,1]$ to $\mathbb{R}^m$. We have to impose constraints on $\theta$ for the model $F_\theta$ to be well-defined. To this aim, we endow (essentially bounded) functions from $[0,1]$ to $\mathbb{R}^m$ with the following $(1,\infty)$-norm

$$\|\theta\|_{1,\infty} = \sup_{0 \leqslant t \leqslant 1} \sum_{i=1}^m |\theta_i(t)|. \tag{6}$$

We can now define the set of parameters

$$\Theta = \{\theta : [0,1] \to \mathbb{R}^m, \|\theta\|_{1,\infty} \leqslant R_\Theta \text{ and } \theta_i \text{ is } K_\Theta\text{-Lipschitz for } i \in \{1,\ldots,m\}\}, \tag{7}$$

for some $R_\Theta > 0$ and $K_\Theta \geqslant 0$. Then, for $\theta \in \Theta$, the following Proposition, which is a consequence of the Picard-Lindelöf Theorem, shows that the mapping $x \mapsto F_\theta(x)$ is well-defined.

**Proposition 1** (Well-posedness of the parameterized ODE). *For $\theta \in \Theta$ and $x \in \mathbb{R}^d$, there exists a unique solution to the initial value problem* (5).

An immediate consequence of Proposition 1 is that it is legitimate to consider $\mathcal{F}_\Theta = \{F_\theta, \theta \in \Theta\}$ for our model class.

When $K_\Theta = 0$, the parameter space $\Theta$ is finite-dimensional since each $\theta_i$ is constant. This setting corresponds to the time-independent neural ODEs of Chen et al. (2018). In this case, the norm (6) reduces to the $\|\cdot\|_1$ norm over $\mathbb{R}^m$. Note that, to fit exactly the formulation of Chen et al. (2018), the time $t$ can be added as a variable of the functions $f_i$, which amounts to adding a new coordinate to $H_t$. This does not change the subsequent analysis. In the richer time-dependent case where $K_\Theta > 0$, the set $\Theta$ belongs to an infinite-dimensional space and therefore, in practice, $\theta_i$ is approximated in a finite basis of functions, such as Fourier series, Chebyshev polynomials, and splines. We refer to Massaroli et al. (2020) for a more detailed discussion, including formulations of the backpropagation algorithm (a.k.a. the adjoint method) in this setting.

Note that we consider the case where the dynamics at time $t$ are linear with respect to the parameter $\theta_i(t)$. Nevertheless, we emphasize that the mapping $x \mapsto F_\theta(x)$ remains a highly non-linear function of each $\theta_i(t)$. To fix ideas, this setting can be seen as analogue to working with pre-activation residual networks instead of post-activation (see He et al., 2016b, for definitions of the terminology), which is a mild modification.

**Statistical analysis**  Since $\Theta$ is a subset of an infinite-dimensional space, complexity measures based on the number of parameters cannot be used. Instead, our approach is to resort to Lipschitz-based complexity measures. More precisely, to bound the complexity of our model class, we propose two building blocks: we first show that the model $F_\theta$ is Lipschitz-continuous with respect to its parameters $\theta$. This allows us to bound the complexity of the model class depending on the complexity of the parameter class. In a second step, we assess the complexity of the class of parameters itself.

Starting with our first step, we show the following estimates for our class of parameterized ODEs. Here and in the following, $\|\cdot\|$ denotes the $\ell_2$ norm over $\mathbb{R}^d$.

**Proposition 2** (The parameterized ODE is bounded and Lipschitz)**.** *Let $\theta$ and $\tilde\theta \in \Theta$. Then, for any $x \in \mathcal{X}$,*

$$\|F_\theta(x)\| \leqslant R_\mathcal{X} + MR_\Theta \exp(K_f R_\Theta)$$

*and*

$$\|F_\theta(x) - F_{\tilde\theta}(x)\| \leqslant 2MK_f R_\Theta \exp(2K_f R_\Theta)\|\theta - \tilde\theta\|_{1,\infty}.$$

The proof, given in the Appendix, makes extensive use of Grönwall's inequality (Pachpatte and Ames, 1997), a standard tool to obtain estimates in the theory of ODEs, in order to bound the magnitude of the solution $H_t$ of (5).

The next step is to assess the magnitude of the covering number of $\Theta$. Recall that, for $\varepsilon > 0$, the $\varepsilon$-covering number of a metric space is the number of balls of radius $\varepsilon$ needed to completely cover the space, with possible overlaps. More formally, considering a metric space $\mathcal{M}$ and denoting by $B(x, \varepsilon)$ the ball of radius $\varepsilon$ centered at $x \in \mathcal{M}$, the $\varepsilon$-covering number of $\mathcal{M}$ is equal to $\inf\{n \geqslant 1 | \exists x_1, \ldots, x_n \in \mathcal{M}, \mathcal{M} \subseteq \bigcup_{i=1}^n B(x_i, \varepsilon)\}$.

**Proposition 3** (Covering number of the ODE parameter class)**.** *For $\varepsilon > 0$, let $\mathcal{N}(\varepsilon)$ be the $\varepsilon$-covering number of $\Theta$ endowed with the distance associated to the $(1, \infty)$-norm (6). Then*

$$\log \mathcal{N}(\varepsilon) \leqslant m \log\left(\frac{16mR_\Theta}{\varepsilon}\right) + \frac{m^2 K_\Theta \log(4)}{\varepsilon}.$$

Proposition 3 is a consequence of a classical result, see, e.g., Kolmogorov and Tikhomirov (1959, example 3 of paragraph 2). A self-contained proof is given in the Appendix for completeness. We also refer to Gottlieb et al. (2017) for more general results on covering numbers of Lipschitz functions.

The two propositions above and an $\varepsilon$-net argument allow to prove the first main result of our paper (where we recall that the notations are defined in Section 3.1).

**Theorem 1** (Generalization bound for parameterized ODEs)**.** *Consider the class of parameterized ODEs $\mathcal{F}_\Theta = \{F_\theta, \theta \in \Theta\}$, where $F_\theta$ is given by (5) and $\Theta$ by (7). Let $\delta > 0$.*

*Then, for $n \geqslant 9\max(m^{-2}R_\Theta^{-2}, 1)$, with probability at least $1 - \delta$,*

$$\mathscr{R}(\widehat\theta_n) \leqslant \widehat{\mathscr{R}}_n(\widehat\theta_n) + B\sqrt{\frac{(m+1)\log(R_\Theta mn)}{n}} + B\frac{m\sqrt{K_\Theta}}{n^{1/4}} + \frac{B}{\sqrt{n}}\sqrt{\log\frac{1}{\delta}},$$

*where $B$ is a constant depending on $K_\ell, K_f, R_\Theta, R_\mathcal{X}, R_\mathcal{Y}, M$. More precisely,*

$$B = 6K_\ell K_f \exp(K_f R_\Theta)\big(R_\mathcal{X} + MR_\Theta \exp(K_f R_\Theta) + R_\mathcal{Y}\big).$$

Three terms appear in our upper bound of $\mathscr{R}(\widehat\theta_n) - \widehat{\mathscr{R}}_n(\widehat\theta_n)$. The first and the third ones are classical (see, e.g. Bach, 2023, Sections 4.4 and 4.5). On the contrary, the second term is more surprising with its convergence rate in $\mathcal{O}(n^{-1/4})$. This slower convergence rate is due to the fact that the space of parameters is infinite-dimensional. In particular, for $K_\Theta = 0$, corresponding to a finite-dimensional space of parameters, we recover the usual $\mathcal{O}(n^{-1/2})$ convergence rate, however at the cost of considering a much more restrictive class of models. Finally, it is noteworthy that the dimensionality appearing in the bound is not the input dimension $d$ but the number of mappings $m$.

Note that this result is general and may be applied in a number of contexts that go beyond deep learning, as long as the instantaneous dependence of the ODE dynamics to the parameters is linear. One such example is the predator-prey model, describing the evolution of two populations of animals, which reads $dx_t = x_t(\alpha - \beta y_t)dt$ and $dy_t = -y_t(\gamma - \delta x_t)dt$, where $x_t$ and $y_t$ are real-valued variables and $\alpha, \beta, \gamma$ and $\delta$ are model parameters. This ODE falls into the framework of this section,

if one were to estimate the parameters by empirical risk minimization. We refer to Deuflhard and Röblitz (2015, section 3) for other examples of parameterized biological ODE dynamics and methods for parameter identification.

Nevertheless, for the sake of brevity, we focus on applications of this result to deep learning, and more precisely to neural ODEs, which is the topic of the next section.

### 3.3 Application to neural ODEs

As explained in Section 1, parameterized ODEs include both time-dependent and time-independent neural ODEs. Since the time-independent model is more common in practice, we develop this case here and leave the time-dependent case to the reader. We thus consider the following neural ODE:

$$
\begin{aligned}
H_0 &= x \\
dH_t &= W\sigma(H_t)dt \\
F_W(x) &= H_1,
\end{aligned}
\tag{8}
$$

where $W \in \mathbb{R}^{d \times d}$ is a weight matrix, and $\sigma : \mathbb{R} \to \mathbb{R}$ is an activation function applied component-wise. We assume $\sigma$ to be $K_\sigma$-Lipschitz for some $K_\sigma > 0$. This assumption is satisfied by all common activation functions. To put the model in the form of Section 3.2, denote $e_1, \ldots, e_d$ the canonical basis of $\mathbb{R}^d$. Then the dynamics (8) can be reformulated as

$$
dH_t = \sum_{i,j=1}^{d} W_{ij}\sigma_{ij}(H_t)dt,
$$

where $\sigma_{ij}(x) = \sigma(x_j)e_i$. Each $\sigma_{ij}$ is itself $K_\sigma$-Lipschitz, hence we fall in the framework of Section 3.2. In other words, the functions $f_i$ of our general parameterized ODE model form a shallow neural network with pre-activation. Denote by $\|W\|_{1,1}$ the sum of the absolute values of the elements of $W$. We consider the following set of parameters, which echoes the set $\Theta$ of Section 3.2:

$$
\mathcal{W} = \{W \in \mathbb{R}^{d \times d}, \|W\|_{1,1} \leqslant R_\mathcal{W}\},
\tag{9}
$$

for some $R_\mathcal{W} > 0$. We can then state the following result as a consequence of Theorem 1.

**Corollary 1** (Generalization bound for neural ODEs). *Consider the class of neural ODEs $\mathcal{F}_\mathcal{W} = \{F_W, W \in \mathcal{W}\}$, where $F_W$ is given by (8) and $\mathcal{W}$ by (9). Let $\delta > 0$.*

*Then, for $n \geqslant 9R_\mathcal{W}^{-1}\max(d^{-4}R_\mathcal{W}^{-1}, 1)$, with probability at least $1 - \delta$,*

$$
\mathscr{R}(\widehat{W}_n) \leqslant \widehat{\mathscr{R}}_n(\widehat{W}_n) + B(d+1)\sqrt{\frac{\log(R_\mathcal{W}dn)}{n}} + \frac{B}{\sqrt{n}}\sqrt{\log\frac{1}{\delta}},
$$

*where $B$ is a constant depending on $K_\ell, K_\sigma, R_\mathcal{W}, R_\mathcal{X}, R_\mathcal{Y}, M$. More precisely,*

$$
B = 6\sqrt{2}K_\ell K_\sigma \exp(K_\sigma R_\mathcal{W})\big(R_\mathcal{X} + MR_\mathcal{W}\exp(K_\sigma R_\mathcal{W}) + R_\mathcal{Y}\big).
$$

Note that the term in $\mathcal{O}(n^{-1/4})$ from Theorem 1 is now absent. Since we consider a time-independent model, we are left with the other two terms, recovering a standard $\mathcal{O}(n^{-1/2})$ convergence rate.

## 4 Generalization bounds for deep residual networks

As highlighted in Section 1, there is a strong connection between neural ODEs and discrete residual neural networks. The previous study of the continuous case in Section 3 paves the way for deriving a generalization bound in the discrete setting of residual neural networks, which is of great interest given the pervasiveness of this architecture in modern deep learning.

We begin by presenting our model and result in Section 4.1, before detailing the comparison of our approach with other papers in Section 4.2 and giving some numerical illustration in Section 4.3.

## 4.1 Model and generalization bound

**Model.** We consider the following class of deep residual networks:

$$H_0 = x$$
$$H_{k+1} = H_k + \frac{1}{L} W_{k+1} \sigma(H_k), \quad 0 \leqslant k \leqslant L - 1 \tag{10}$$
$$F_{\mathbf{W}}(x) = H_L,$$

where the parameter $\mathbf{W} = (W_k)_{1 \leqslant k \leqslant L} \in \mathbb{R}^{L \times d \times d}$ is a set of weight matrices and $\sigma$ is still a $K_\sigma$-Lipschitz activation function. To emphasize that $\mathbf{W}$ is here a third-order tensor, as opposed to the case of time-invariant neural ODEs in Section 3.3, where $W$ was a matrix, we denote it with a bold notation. We also assume in the following that $\sigma(0) = 0$. This assumption could be alleviated at the cost of additional technicalities. Owing to the $1/L$ scaling factor, the deep limit of this residual network is a (time-dependent) neural ODE of the form studied in Section 3. We refer to Marion et al. (2022) for further discussion on the link between scaling factors and deep limits. We simply note that this scaling factor is not common practice, but preliminary experiments show it does not hurt performance and can even improve performance in a weight-tied setting (Sander et al., 2022). The space of parameters is endowed with the following $(1, 1, \infty)$-norm

$$\|\mathbf{W}\|_{1,1,\infty} = \sup_{1 \leqslant k \leqslant L} \sum_{i,j=1}^{d} |W_{k,i,j}|. \tag{11}$$

Also denoting $\|\cdot\|_\infty$ the element-wise maximum norm for a matrix, we consider the class of matrices

$$\mathcal{W} = \left\{ \mathbf{W} \in \mathbb{R}^{L \times d \times d}, \quad \|\mathbf{W}\|_{1,1,\infty} \leqslant R_\mathcal{W} \quad \text{and} \right.$$
$$\left. \|W_{k+1} - W_k\|_\infty \leqslant \frac{K_\mathcal{W}}{L} \text{ for } 1 \leqslant k \leqslant L - 1 \right\}, \tag{12}$$

for some $R_\mathcal{W} > 0$ and $K_\mathcal{W} \geqslant 0$, which is a discrete analogue of the set $\Theta$ defined by (7).

In particular, the upper bound on the difference between successive weight matrices is to our knowledge a novel way of constraining the parameters of a neural network. It corresponds to the discretization of the Lipschitz continuity of the parameters introduced in (7). By analogy, we refer to it as a constraint on the Lipschitz constant of the weights. Note that, for standard initialization schemes, the difference between two successive matrices is of the order $\mathcal{O}(1)$ and not $\mathcal{O}(1/L)$, or, in other words, $K_\mathcal{W}$ scales as $\mathcal{O}(L)$. This dependence of $K_\mathcal{W}$ on $L$ can be lifted by adding correlations across layers at initialization. For instance, one can take, for $k \in \{1, \dots, L\}$ and $i, j \in \{1, \dots, d\}$, $\mathbf{W}_{k,i,j} = \frac{1}{\sqrt{d}} f_{i,j}(\frac{k}{L})$, where $f_{i,j}$ is a smooth function, for example a Gaussian process with the RBF kernel. Such a non-i.i.d. initialization scheme is necessary for the correspondence between deep residual networks and neural ODEs to hold (Marion et al., 2022). Furthermore, Sander et al. (2022) prove that, with this initialization scheme, the constraint on the Lipschitz constant also holds for the *trained* network, with $K_\mathcal{W}$ independent of $L$. Finally, we emphasize that the following developments also hold in the case where $K_\mathcal{W}$ depends on $L$ (see also Section 4.2 for a related discussion).

**Statistical analysis.** At first sight, a reasonable strategy would be to bound the distance between the model (10) and its limit $L \to \infty$ that is a parameterized ODE, then *apply* Theorem 1. This strategy is straightforward, but comes at the cost of an additional $\mathcal{O}(1/L)$ term in the generalization bound, as a consequence of the discretization error between the discrete iterations (10) and their continuous limit. For example, we refer to Fermanian et al. (2021) where this strategy is used to prove a generalization bound for discrete RNNs and where this additional error term is incurred. We follow another way by mimicking all the proof with a finite $L$. This is a longer approach but it yields a sharper result since we avoid the $\mathcal{O}(1/L)$ discretization error. The proof structure is similar to Section 3: the following two Propositions are the discrete counterparts of Propositions 2 and 3.

**Proposition 4** (The residual network is bounded and Lipschitz). *Let $\mathbf{W}$ and $\tilde{\mathbf{W}} \in \mathcal{W}$. Then, for any $x \in \mathcal{X}$,*

$$\|F_{\mathbf{W}}(x)\| \leqslant R_\mathcal{X} \exp(K_\sigma R_\mathcal{W})$$

*and*

$$\|F_{\mathbf{W}}(x) - F_{\tilde{\mathbf{W}}}(x)\| \leqslant \frac{R_\mathcal{X}}{R_\mathcal{W}} \exp(2K_\sigma R_\mathcal{W}) \|\mathbf{W} - \tilde{\mathbf{W}}\|_{1,1,\infty}.$$

**Proposition 5** (Covering number of the residual network parameter class)**.** *Let $\mathcal{N}(\varepsilon)$ be the covering number of $\mathcal{W}$ endowed with the distance associated to the $(1, 1, \infty)$-norm* (11)*. Then*

$$\log \mathcal{N}(\varepsilon) \leqslant d^2 \log \Big( \frac{16 d^2 R_{\mathcal{W}}}{\varepsilon} \Big) + \frac{d^4 K_{\mathcal{W}} \log(4)}{\varepsilon}.$$

The proof of Proposition 4 is a discrete analogous of Proposition 2. On the other hand, Proposition 5 can be proven as a *consequence* of Proposition 3, by showing the existence of an injective isometry from $\mathcal{W}$ into a set of the form (7). Equipped with these two propositions, we are now ready to state the generalization bound for our class of residual neural networks.

**Theorem 2** (Generalization bound for deep residual networks)**.** *Consider the class of neural networks $\mathcal{F}_{\mathcal{W}} = \{F_{\mathbf{W}}, \mathbf{W} \in \mathcal{W}\}$, where $F_{\mathbf{W}}$ is given by* (10) *and $\mathcal{W}$ by* (12)*. Let $\delta > 0$.*

*Then, for $n \geqslant 9 R_{\mathcal{W}}^{-1} \max(d^{-4} R_{\mathcal{W}}^{-1}, 1)$, with probability at least $1 - \delta$,*

$$\mathscr{R}(\widehat{\mathbf{W}}_n) \leqslant \widehat{\mathscr{R}}_n(\widehat{\mathbf{W}}_n) + B(d+1) \sqrt{\frac{\log(R_{\mathcal{W}} dn)}{n}} + B \frac{d^2 \sqrt{K_{\mathcal{W}}}}{n^{1/4}} + \frac{B}{\sqrt{n}} \sqrt{\log \frac{1}{\delta}}, \qquad (13)$$

*where $B$ is a constant depending on $K_\ell, K_\sigma, R_{\mathcal{W}}, R_{\mathcal{X}}, R_{\mathcal{Y}}$. More precisely,*

$$B = 6\sqrt{2} K_\ell \max \Big( \frac{\exp(K_\sigma R_{\mathcal{W}})}{R_{\mathcal{W}}}, 1 \Big) (R_{\mathcal{X}} \exp(K_\sigma R_{\mathcal{W}}) + R_{\mathcal{Y}}).$$

We emphasize that this result is non-asymptotic and valid for any width $d$ and depth $L$. Furthermore, the depth $L$ does not appear in the upper bound (13). This should not surprise the reader since Theorem 1 can be seen as the deep limit $L \to \infty$ of this result, hence we expect that our bound remains finite when $L \to \infty$ (otherwise the bound of Theorem 1 would be infinite). However, $L$ appears as a scaling factor in the definition of the neural network (10) and of the class of parameters (12). This is crucial for the depth independence to hold, as we will comment further on in the next section.

Furthermore, the depth independence comes at the price of a $\mathcal{O}(n^{-1/4})$ convergence rate. Note that, by taking $K_{\mathcal{W}} = 0$, we obtain a generalization bound for weight-tied neural networks with a faster convergence rate in $n$, since the term in $\mathcal{O}(n^{-1/4})$ vanishes.

## 4.2 Comparison with other bounds

As announced in Section 2, we now compare Theorem 2 with the results of Bartlett et al. (2017) and Golowich et al. (2018). Beginning by Bartlett et al. (2017), we first state a slightly weaker version of their result to match our notations and facilitate comparison.

**Corollary 2** (corollary of Theorem 1.1 of Bartlett et al. (2017))**.** *Consider the class of neural networks $\mathcal{F}_{\tilde{\mathcal{W}}} = \{F_{\mathbf{W}}, \mathbf{W} \in \tilde{\mathcal{W}}\}$, where $F_{\mathbf{W}}$ is given by* (10) *and $\tilde{\mathcal{W}} = \{\mathbf{W} \in \mathbb{R}^{L \times d \times d}, \|\mathbf{W}\|_{1,1,\infty} \leqslant R_{\mathcal{W}}\}$.*

*Assume that $L \geqslant R_{\mathcal{W}}$ and $K_\sigma = 1$, and let $\gamma, \delta > 0$. Consider $(x, y), (x_1, y_1), \dots, (x_n, y_n)$ drawn i.i.d. from any probability distribution over $\mathbb{R}^d \times \{1, \dots d\}$ such that a.s. $\|x\| \leqslant R_{\mathcal{X}}$.*

*Then, with probability at least $1 - \delta$, for every $\mathbf{W} \in \tilde{\mathcal{W}}$,*

$$\mathbb{P} \Big( \arg\max_{1 \leqslant j \leqslant d} F_{\mathbf{W}}(x)_j \neq y \Big) \leqslant \widehat{\mathscr{R}}_n(\mathbf{W}) + C \frac{R_{\mathcal{X}} R_{\mathcal{W}} \exp(R_{\mathcal{W}}) \log(d) \sqrt{L}}{\gamma \sqrt{n}} + \frac{C}{\sqrt{n}} \sqrt{\log \frac{1}{\delta}}, \quad (14)$$

*where $\widehat{\mathscr{R}}_n(\mathbf{W}) \leqslant n^{-1} \sum_{i=1}^n \mathbf{1}_{F_{\mathbf{W}}(x_i)_{y_i} \leqslant \gamma + \max_{j \neq y_i} f(x_i)_j}$ and $C$ is a universal constant.*

We first note that the setting is slightly different from ours: they consider a large margin predictor for a multi-class classification problem, whereas we consider a general Lipschitz-continuous loss $\ell$. This being said, the model class is identical to ours, except for one notable difference: the constraint on the Lipschitz constant of the weights appearing in equation (12) is not required here.

Comparing (13) and (14), we see that our bound enjoys a better dependence on the depth $L$ but a worse dependence on the width $d$. Regarding the depth, our bound (13) does not depend on $L$, whereas the bound (14) scales as $\mathcal{O}(\sqrt{L})$. This comes from the fact that we consider a smaller set of parameters (12), by adding the constraint on the Lipschitz norm of the weights. This constraint

allows us to control the complexity of our class of neural networks independently of depth, as long as $K_{\mathcal{W}}$ is independent of $L$. If $K_{\mathcal{W}}$ scales as $\mathcal{O}(L)$, which is the case for i.i.d. initialization schemes, our result also features a scaling in $\mathcal{O}(\sqrt{L})$. As for the width, Bartlett et al. (2017) achieve a better dependence by a subtle covering numbers argument that takes into account the geometry induced by matrix norms. Since our paper focuses on a depth-wise analysis by leveraging the similarity between residual networks and their infinite-depth counterpart, improving the scaling of our bound with width is left for future work. Finally, note that both bounds have a similar exponential dependence in $R_{\mathcal{W}}$.

As for Golowich et al. (2018), they consider non-residual neural networks of the form $x \mapsto M_L \sigma(M_{L-1} \sigma(\ldots \sigma(M_1 x)))$. These authors show that the generalization error of this class scales as

$$ \mathcal{O}\bigg( R_{\mathcal{X}} \frac{\Pi_F \sqrt{\log\big(\frac{\Pi_F}{\pi_S}\big)}}{n^{1/4}} \bigg), $$

where $\Pi_F$ is an upper-bound on the product of the Frobenius norms $\prod_{k=1}^{L} \|M_k\|_F$ and $\pi_S$ is a lower-bound on the product of the spectral norms $\prod_{k=1}^{L} \|M_k\|$. Under the assumption that both $\Pi_F$ and $\Pi_F/\pi_S$ are bounded independently of $L$, their bound is indeed depth-independent, similarly to ours. Interestingly, as ours, the bound presents a $\mathcal{O}(n^{-1/4})$ convergence rate instead of the more usual $\mathcal{O}(n^{-1/2})$. However, the assumption that $\Pi_F$ is bounded independently of $L$ does not hold in our residual setting, since we have $M_k = I + \frac{1}{L}W_k$ and thus we can lower-bound

$$ \prod_{k=1}^{L} \|M_k\|_F \geqslant \prod_{k=1}^{L} \Big( \|I\|_F - \frac{1}{L}\|M_k\|_F \Big) \geqslant \big(\sqrt{d} - \frac{R_{\mathcal{W}}}{L}\big)^L \approx d^{\frac{L}{2}} e^{-\frac{R_{\mathcal{W}}}{\sqrt{d}}}. $$

In our setting, it is a totally different assumption, the constraint that two successive weight matrices should be close to one another, which allows us to derive depth-independent bounds.

### 4.3 Numerical illustration

The bound of Theorem 2 features two quantities that depend on the class of neural networks, namely $R_{\mathcal{W}}$ that bounds a norm of the weight matrices and $K_{\mathcal{W}}$ that bounds the maximum *difference* between two successive weight matrices, i.e. the Lipschitz constant of the weights. The first one belongs to the larger class of norm-based bounds that has been extensively studied (see, e.g., Neyshabur et al., 2015). We are therefore interested in getting a better understanding of the role of the second quantity, which is much less common, in the generalization ability of deep residual networks.

To this aim, we train deep residual networks (10) (of width $d = 30$ and depth $L = 1000$) on MNIST. We prepend the network with an initial weight matrix to project the data $x$ from dimension 768 to dimension 30, and similarly postpend it with another matrix to project the output $F_{\mathbf{W}}(x)$ into dimension 10 (i.e. the number of classes in MNIST). Finally, we consider two training settings: either the initial and final matrices are trained, or they are fixed random projections. We use the initialization scheme outlined in Section 4.1. Further experimental details are postponed to the Appendix.

We report in Figure 1a the generalization gap of the trained networks, that is, the difference between the test and train errors (in terms of cross entropy loss), as a function of the maximum Lipschitz constant of the weights $\sup_{0 \leqslant k \leqslant L-1}(\|W_{k+1} - W_k\|_\infty)$. We observe a positive correlation between these two quantities. To further analyze the relationship between the Lipschitz constant of the weights and the generalization gap, we then add the penalization term $\lambda \cdot \big( \sum_{k=0}^{L-1} \|W_{k+1} - W_k\|_F^2 \big)^{1/2}$ to the loss, for some $\lambda \geqslant 0$. The obtained generalization gap is reported in Figure 1b as a function of $\lambda$. We observe that this penalization allows to reduce the generalization gap. These two observations go in support of the fact that a smaller Lipschitz constant improves the generalization power of deep residual networks, in accordance with Theorem 2.

However, note that we were not able to obtain an improvement on the test loss by adding the penalization term. This is not all too surprising since previous work has investigated a related penalization, in terms of the Lipschitz norm of the layer sequence $(H_k)_{0 \leqslant k \leqslant L}$, and was similarly not able to report any improvement on the test loss (Kelly et al., 2020).

Finally, the proposed penalization term slightly departs from the theory that involves $\sup_{0 \leqslant k \leqslant L-1}(\|W_{k+1} - W_k\|_\infty)$. This is because the maximum norm is too irregular to be used

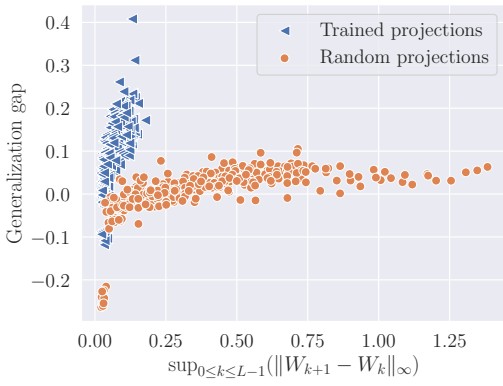 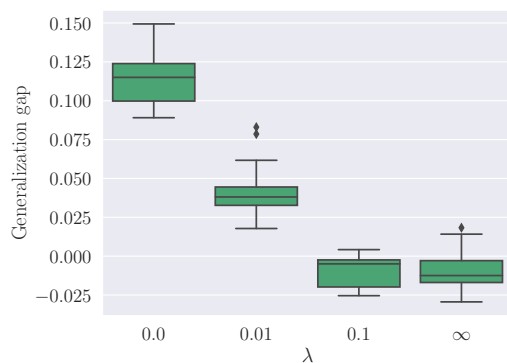

(a) Generalization gap as a function of the maximum Lipschitz constant of the weights. Each dot corresponds to a network trained with a varying number of epochs (between 1 and 30).

(b) Generalization gap as a function of the penalization factor $\lambda$. The experiment is repeated 20 times for each value of $\lambda$. Each time, the network is trained for 50 epochs. The initial and final matrices are random. The value $\lambda = \infty$ corresponds to a weight-tied network.

Figure 1: Link between the generalization gap and the Lipschitz constant of the weights.

in practice since, at any one step of gradient descent, it only impacts the maximum weights and not the others. As an illustration, Figure 2 shows the generalization gap when penalizing with the maximum max-norm $\sup_{0 \leqslant k \leqslant L-1}(\|W_{k+1} - W_k\|_\infty)$ and the $L_2$ norm of the max-norm $\left(\sum_{k=0}^{L-1} \|W_{k+1} - W_k\|_\infty^2\right)^{1/2}$. The factor $\lambda$ is scaled appropriately to reflect the scale difference of the penalizations. The results are mixed: the $L_2$ norm of the max-norm is effective contrarily to the maximum max-norm. Further investigation of the properties of these norms is left for future work.

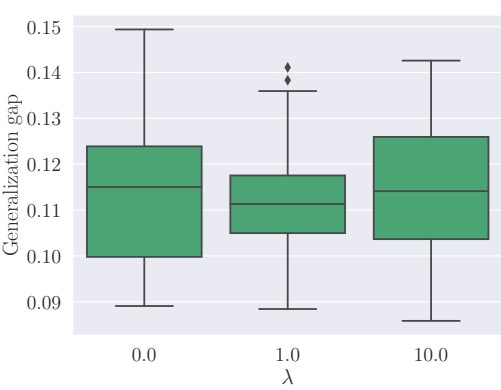 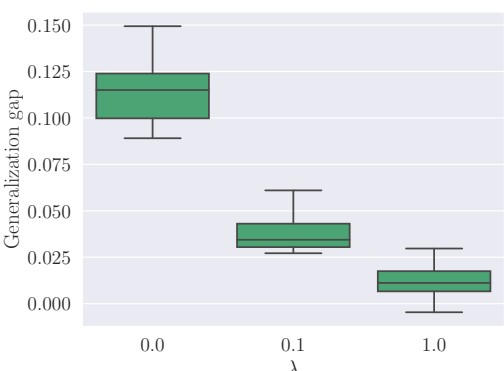

(a) Penalizing by the maximum max-norm.

(b) Penalizing by the $L_2$ norm of the max-norm.

Figure 2: Generalization gap as a function of the penalization factor $\lambda$ for other penalizations.

## 5  Conclusion

We provide a generalization bound that applies to a wide range of parameterized ODEs. As a consequence, we obtain the first generalization bounds for time-independent and time-dependent neural ODEs in supervised learning tasks. By discretizing our reasoning, we also provide a bound for a class of deep residual networks. Understanding the approximation and optimization properties of this class of neural networks is left for future work. Another intriguing extension is to relax the assumption of linearity of the dynamics at time $t$ with respect to $\theta_i(t)$, that is, to consider a general formulation $dH_t = \sum_{i=1}^m f_i(H_t, \theta_i(t))$. In the future, it should also be interesting to extend our results to the more involved case of neural SDEs, which have also been found to be deep limits of a large class of residual neural networks (Cohen et al., 2021; Marion et al., 2022).

## Acknowledgments and Disclosure of Funding

The author is supported by a grant from Région Île-de-France, by a Google PhD Fellowship, and by MINES Paris - PSL. The author thanks Eloïse Berthier, Gérard Biau, Adeline Fermanian, Clément Mantoux, and Jean-Philippe Vert for inspiring discussions, thorough proofreading and suggestions on this paper.

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

# Appendix

**Organization of the Appendix** Section A contains the proofs of the results of the main paper. Section B contains the details of the numerical illustrations presented in Section 4.3.

## A  Proofs

### A.1  Proof of Proposition 1

The function

$$(t, h) \mapsto \sum_{i=1}^{m} \theta_i(t) f_i(h)$$

is locally Lipschitz-continuous with respect to its first variable and globally Lipschitz-continuous with respect to its second variable. Therefore, the existence and uniqueness of the solution of the initial value problem (5) for $t \geqslant 0$ comes as a consequence of the Picard-Lindelöf theorem (see, e.g., Luk, 2017 for a self-contained presentation and Arnold, 1992 for a textbook).

### A.2  Proof of Proposition 2

For $x \in \mathcal{X}$, let $H$ be the solution of the initial value problem (5) with parameter $\theta$ and with the initial condition $H_0 = x$. Let us first upper-bound $\|f_i(H_t)\|$ for all $i \in \{1, \ldots, m\}$ and $t > 0$. To this aim, for $t \geqslant 0$, we have

$$
\begin{aligned}
\|H_t - H_0\| = \left\| \int_0^t \sum_{i=1}^{m} \theta_i(s) f_i(H_s) ds \right\| \\
\leqslant \int_0^t \sum_{i=1}^{m} |\theta_i(s)| \|f_i(H_0)\| ds + \int_0^t \sum_{i=1}^{m} |\theta_i(s)| \|f_i(H_s) - f_i(H_0)\| ds \\
\leqslant M \int_0^t \sum_{i=1}^{m} |\theta_i(s)| ds + K_f \int_0^t \left( \|H_s - H_0\| \sum_{i=1}^{m} |\theta_i(s)| \right) ds \\
\leqslant t M R_\Theta + K_f R_\Theta \int_0^t \|H_s - H_0\| ds.
\end{aligned}
$$

Next, Grönwall's inequality yields, for $t \in [0, 1]$,

$$\|H_t - H_0\| \leqslant t M R_\Theta \exp(t K_f R_\Theta) \leqslant M R_\Theta \exp(K_f R_\Theta).$$

Hence

$$\|H_t\| \leqslant \|H_0\| + \|H_t - H_0\| \leqslant R_\mathcal{X} + M R_\Theta \exp(K_f R_\Theta),$$

yielding the first result of the proposition. Furthermore, for any $i \in \{1, \ldots, m\}$,

$$\|f_i(H_t)\| \leqslant \|f_i(H_t) - f_i(H_0)\| + \|f_i(H_0)\| \leqslant M\big(K_f R_\Theta \exp(K_f R_\Theta) + 1\big) =: C.$$

Now, let $\tilde{H}$ be the solution of the initial value problem (5) with another parameter $\tilde{\theta}$ and with the same initial condition $\tilde{H}_0 = x$. Then, for any $t \geqslant 0$,

$$H_t - \tilde{H}_t = \int_0^t \sum_{i=1}^{m} \theta_i(s) f_i(H_s) ds - \int_0^t \sum_{i=1}^{m} \tilde{\theta}_i(s) f_i(\tilde{H}_s) ds.$$

Hence

$$\|H_t - \tilde{H}_t\| = \Big\| \int_0^t \sum_{i=1}^m (\theta_i(s) - \tilde{\theta}_i(s)) f_i(H_s) ds + \int_0^t \sum_{i=1}^m \tilde{\theta}_i(s)(f_i(H_s) - f_i(\tilde{H}_s)) ds \Big\|$$

$$\leqslant \int_0^t \sum_{i=1}^m |\theta_i(s) - \tilde{\theta}_i(s)| \|f_i(H_s)\| ds + \int_0^t \sum_{i=1}^m |\tilde{\theta}_i(s)| \|f_i(H_s) - f_i(\tilde{H}_s)\| ds$$

$$\leqslant \int_0^t \sum_{i=1}^m |\theta_i(s) - \tilde{\theta}_i(s)| \|f_i(H_s)\| ds + K_f \int_0^t \Big( \|H_s - \tilde{H}_s\| \sum_{i=1}^m |\tilde{\theta}_i(s)| \Big) ds$$

$$\leqslant tC\|\theta - \tilde{\theta}\|_{1,\infty} + K_f R_\Theta \int_0^t \|H_s - \tilde{H}_s\| ds.$$

Then Grönwall's inequality implies that, for $t \in [0,1]$,

$$\|H_t - \tilde{H}_t\| \leqslant tC\|\theta - \tilde{\theta}\|_{1,\infty} \exp(tK_f R_\Theta)$$

$$\leqslant M(K_f R_\Theta \exp(K_f R_\Theta) + 1) \exp(K_f R_\Theta) \|\theta - \tilde{\theta}\|_{1,\infty}$$

$$\leqslant 2M K_f R_\Theta \exp(2K_f R_\Theta) \|\theta - \tilde{\theta}\|_{1,\infty}$$

since $1 \leqslant K_f R_\Theta \exp(K_f R_\Theta)$ because $K_f \geqslant 1$, $R_\Theta \geqslant 1$.

### A.3 Proof of Proposition 3

We first prove the result for $m = 1$. Let $G_x$ be an $\varepsilon/2K_\Theta$-grid of $[0,1]$ and $G_y$ an $\varepsilon/2$-grid of $[-R_\Theta, R_\Theta]$. Formally, we can take

$$G_x = \Big\{ \frac{k\varepsilon}{2K_\Theta}, \, 0 \leqslant k \leqslant \Big\lceil \frac{2K_\Theta}{\varepsilon} \Big\rceil \Big\} \quad \text{and} \quad G_y = \Big\{ -R_\Theta + \frac{k\varepsilon}{2}, \, 1 \leqslant k \leqslant \Big\lfloor \frac{4R_\Theta}{\varepsilon} \Big\rfloor \Big\}$$

Our cover consists of all functions that start at a point of $G_y$, are piecewise linear with kinks in $G_x$, where each piece has slope $+K_\Theta$ or $-K_\Theta$. Hence our cover is of size

$$\mathcal{N}_1(\varepsilon) = |G_y| 2^{|G_x|} \leqslant \frac{4R_\Theta}{\varepsilon} 2^{\frac{2K_\Theta}{\varepsilon}+2} = \frac{16R_\Theta}{\varepsilon} 4^{\frac{K_\Theta}{\varepsilon}}.$$

Now take a function $f : [0,1] \to \mathbb{R}$ that is uniformly bounded by $R_\Theta$ and $K_\Theta$-Lipschitz. We construct a cover member at distance $\varepsilon$ from $f$ as follows. Choose a point $y_0$ in $G_y$ at distance at most $\varepsilon/2$ from $f(0)$. Since $f(0) \in [-R_\Theta, R_\Theta]$, this is clearly possible, except perhaps at the end of the interval. To verify that it is possible at the end of the interval, note that $R_\Theta$ is at a distance less than $\varepsilon/2$ of the last element of the grid, since

$$R_\Theta - \Big( -R_\Theta + \Big\lfloor \frac{4R_\Theta}{\varepsilon} \Big\rfloor \frac{\varepsilon}{2} \Big) = 2R_\Theta - \Big\lfloor \frac{4R_\Theta}{\varepsilon} \Big\rfloor \frac{\varepsilon}{2} \in \Big[ 2R_\Theta - \frac{4R_\Theta}{\varepsilon}\frac{\varepsilon}{2}, 2R_\Theta - \Big(\frac{4R_\Theta}{\varepsilon} - 1\Big)\frac{\varepsilon}{2} \Big] = \Big[ 0, \frac{\varepsilon}{2} \Big].$$

Then, among the cover members that start at $y_0$, choose the one which is closest to $f$ at each point of $G_x$ (in case of equality, pick any one). Let us denote this cover member as $\tilde{f}$. Let us show recursively that $f$ is at $\ell_\infty$-distance at most $\varepsilon$ from $\tilde{f}$. More precisely, let us first show by induction on $k$ that for all $k \in \{0, \ldots, \lceil \frac{2K_\Theta}{\varepsilon} \rceil\}$,

$$\Big| f\Big(\frac{k\varepsilon}{2K_\Theta}\Big) - \tilde{f}\Big(\frac{k\varepsilon}{2K_\Theta}\Big) \Big| \leqslant \frac{\varepsilon}{2}. \tag{15}$$

First, $|f(0) - \tilde{f}(0)| \leqslant \frac{\varepsilon}{2}$. Then, assume that (15) holds for some $k$. Then we have the following inequalities:

$$\tilde{f}\Big(\frac{k\varepsilon}{2K_\Theta}\Big) - \varepsilon \leqslant f\Big(\frac{k\varepsilon}{2K_\Theta}\Big) - \frac{\varepsilon}{2} \qquad \text{(by induction)}$$

$$\leqslant f\Big(\frac{(k+1)\varepsilon}{2K_\Theta}\Big) \qquad (f \text{ is } K_\Theta\text{-Lipschitz})$$

$$\leqslant f\Big(\frac{k\varepsilon}{2K_\Theta}\Big) + \frac{\varepsilon}{2} \qquad (f \text{ is } K_\Theta\text{-Lipschitz})$$

$$\leqslant \tilde{f}\Big(\frac{k\varepsilon}{2K_\Theta}\Big) + \varepsilon \qquad \text{(by induction)}.$$

Moreover, by definition, $\tilde{f}\big(\frac{(k+1)\varepsilon}{K_\Theta}\big)$ is the closest point to $f\big(\frac{(k+1)\varepsilon}{K_\Theta}\big)$ among

$$\left\{\tilde{f}\big(\frac{k\varepsilon}{K_\Theta}\big) - \frac{\varepsilon}{2}, \tilde{f}\big(\frac{k\varepsilon}{K_\Theta}\big) + \frac{\varepsilon}{2}\right\}.$$

The bounds above show that, among those two points, at least one is at distance no more than $\varepsilon/2$ from $f\big(\frac{(k+1)\varepsilon}{K_\Theta}\big)$. This shows (15) at rank $k+1$.

To conclude, take now $x \in [0,1]$. There exists $k \in \{0, \ldots, \lceil \frac{2K_\Theta}{\varepsilon}\rceil\}$ such that $x$ is at distance at most $\varepsilon/4K_\Theta$ from $\frac{k\varepsilon}{2K_\Theta}$. Again, this is clear except perhaps at the end of the interval, where it is also true since

$$1 - \left\lceil \frac{2K_\Theta}{\varepsilon}\right\rceil \frac{\varepsilon}{2K_\Theta} \leqslant 1 - \frac{2K_\Theta}{\varepsilon}\frac{\varepsilon}{2K_\Theta} = 0,$$

meaning that 1 is located between two elements of the grid $G_x$, showing that it is at distance at most $\varepsilon/4K_\Theta$ from one element of the grid. Then, we have

$$|f(x) - \tilde{f}(x)| \leqslant \left|f(x) - f\big(\frac{k\varepsilon}{2K_\Theta}\big)\right| + \left|f\big(\frac{k\varepsilon}{2K_\Theta}\big) - \tilde{f}\big(\frac{k\varepsilon}{2K_\Theta}\big)\right| + \left|\tilde{f}\big(\frac{k\varepsilon}{2K_\Theta}\big) - \tilde{f}(x)\right|$$
$$\leqslant \frac{\varepsilon}{4} + \frac{\varepsilon}{2} + \frac{\varepsilon}{4},$$

where the first and third terms are upper-bounded because $f$ and $\tilde{f}$ are $K_\Theta$-Lip, while the second term is upper bounded by (15). Hence $\|f - \tilde{f}\|_\infty \leqslant \varepsilon$, proving the result for $m = 1$.

Finally, to prove the result for a general $m$, note that the Cartesian product of $\varepsilon/m$-covers for each coordinate of $\theta$ gives an $\varepsilon$-cover for $\theta$. Indeed, consider such covers and take $\theta \in \Theta$. Since each coordinate of $\theta$ is uniformly bounded by $R_\Theta$ and $K_\Theta$-Lipschitz, the proof above shows the existence of a cover member $\tilde{\theta}$ such that, for all $i \in \{1, \ldots, m\}$, $\|\theta_i - \tilde{\theta}_i\|_\infty \leqslant \varepsilon/m$. Then

$$\|\theta - \tilde{\theta}\|_{1,\infty} = \sup_{0 \leqslant t \leqslant 1} \sum_{i=1}^m |\theta_i(t) - \tilde{\theta}_i(t)| \leqslant \sup_{0 \leqslant t \leqslant 1} \sum_{i=1}^m \|\theta_i - \tilde{\theta}_i\|_\infty \leqslant \varepsilon.$$

As a consequence, we conclude that

$$\mathcal{N}(\varepsilon) \leqslant \left(\mathcal{N}_1\big(\frac{\varepsilon}{m}\big)\right)^m = \left(\frac{16mR_\Theta}{\varepsilon}\right)^m 4^{\frac{m^2 K_\Theta}{\varepsilon}}.$$

Taking the logarithm yields the result.

### A.4  Proof of Theorem 1

First note that, for any $\theta \in \Theta$, $x \in \mathcal{X}$ and $y \in \mathcal{Y}$,

$$|\ell(F_\theta(x), y)| \leqslant |\ell(F_\theta(x), y) - \ell(y, y)| + |\ell(y, y)| \leqslant K_\ell \|F_\theta(x) - y\|.$$

since, by assumption, $\ell$ is $K_\ell$-Lipschitz with respect to its first variable and $\ell(y, y) = 0$. Thus

$$|\ell(F_\theta(x), y)| \leqslant K_\ell \big(\|F_\theta(x)\| + \|y\|\big) \leqslant K_\ell\big(R_\mathcal{X} + MR_\Theta \exp(K_f R_\Theta) + R_\mathcal{Y}\big) =: \overline{M}$$

by Proposition 2.

Now, taking $\delta > 0$, a classical computation involving McDiarmid's inequality (see, e.g., Wainwright, 2019, proof of thm 4.10) yields that, with probability at least $1 - \delta$,

$$\mathcal{R}(\widehat{\theta}_n) \leqslant \widehat{\mathcal{R}}_n(\widehat{\theta}_n) + \mathbb{E}\Big[\sup_{\theta \in \Theta} |\mathcal{R}(\theta) - \widehat{\mathcal{R}}_n(\theta)|\Big] + \frac{\overline{M}\sqrt{2}}{\sqrt{n}}\sqrt{\log\frac{1}{\delta}}.$$

Denote $C = 2MK_f R_\Theta \exp(2K_f R_\Theta)$. Then we show that $\mathcal{R}$ and $\widehat{\mathcal{R}}_n$ are $CK_\ell$-Lipschitz with respect to $(\theta, \|\cdot\|_{1,\infty})$: for $\theta, \tilde{\theta} \in \Theta$,

$$|\mathcal{R}(\theta) - \mathcal{R}(\tilde{\theta})| \leqslant \mathbb{E}\big[|\ell(F_\theta(x), y) - \ell(F_{\tilde{\theta}}(x), y)|\big]$$
$$\leqslant K_\ell \mathbb{E}\big[\|F_\theta(x) - F_{\tilde{\theta}}(x)\|\big]$$
$$\leqslant CK_\ell \|\theta - \tilde{\theta}\|_{1,\infty},$$

according to Proposition 2. The proof for the empirical risk is very similar.

Let now $\varepsilon > 0$ and $\mathcal{N}(\varepsilon)$ be the covering number of $\Theta$ endowed with the $(1, \infty)$-norm. By Proposition 3,

$$\log \mathcal{N}(\varepsilon) \leqslant m \log \left( \frac{16 m R_\Theta}{\varepsilon} \right) + \frac{m^2 K_\Theta \log(4)}{\varepsilon}.$$

Take $\theta^{(1)}, \ldots, \theta^{(\mathcal{N}(\varepsilon))}$ the associated cover elements. Then, for any $\theta \in \Theta$, denoting $\theta^{(i)}$ the cover element at distance at most $\varepsilon$ from $\theta$,

$$|\mathcal{R}(\theta) - \widehat{\mathcal{R}}_n(\theta)| \leqslant |\mathcal{R}(\theta) - \mathcal{R}(\theta^{(i)})| + |\mathcal{R}(\theta^{(i)}) - \widehat{\mathcal{R}}_n(\theta^{(i)})| + |\widehat{\mathcal{R}}_n(\theta^{(i)}) - \widehat{\mathcal{R}}_n(\theta)|$$
$$\leqslant 2 C K_\ell \varepsilon + \sup_{i \in \{1, \ldots, \mathcal{N}(\varepsilon)\}} |\mathcal{R}(\theta^{(i)}) - \widehat{\mathcal{R}}_n(\theta^{(i)})|.$$

Hence

$$\mathbb{E}\left[ \sup_{\theta \in \Theta} |\mathcal{R}(\theta) - \widehat{\mathcal{R}}_n(\theta)| \right] \leqslant 2 C K_\ell \varepsilon + \mathbb{E}\left[ \sup_{i \in \{1, \ldots, \mathcal{N}(\varepsilon)\}} |\mathcal{R}(\theta^{(i)}) - \widehat{\mathcal{R}}_n(\theta^{(i)})| \right].$$

Recall that a real-valued random variable $X$ is said to be $s^2$ sub-Gaussian (Bach, 2023, Section 1.2.1) if for all $\lambda \in \mathbb{R}$, $\mathbb{E}(\exp(\lambda(X - \mathbb{E}(X)))) \leqslant \exp(\lambda^2 s^2/2)$. Since $\widehat{\mathcal{R}}_n(\theta)$ is the average of $n$ independent random variables, which are each almost surely bounded by $\overline{M}$, it is $\overline{M}^2/n$ sub-Gaussian, hence we have the classical inequality on the expectation of the maximum of sub-Gaussian random variables (see, e.g., Bach, 2023, Exercise 1.13)

$$\mathbb{E}\left[ \sup_{i \in \{1, \ldots, \mathcal{N}(\varepsilon)\}} |\mathcal{R}(\theta^{(i)}) - \widehat{\mathcal{R}}_n(\theta^{(i)})| \right] \leqslant \overline{M} \sqrt{\frac{2 \log(2 \mathcal{N}(\varepsilon))}{n}}.$$

The remainder of the proof consists in computations to put the result in the required format. More precisely, we have

$$\mathbb{E}\left[ \sup_{\theta \in \Theta} |\mathcal{R}(\theta) - \widehat{\mathcal{R}}_n(\theta)| \right] \leqslant 2 C K_\ell \varepsilon + \overline{M} \sqrt{\frac{2 \log(2 \mathcal{N}(\varepsilon))}{n}}$$

$$\leqslant 2 C K_\ell \varepsilon + \overline{M} \sqrt{\frac{2 \log(2) + 2m \log \left( \frac{16 m R_\Theta}{\varepsilon} \right) + \frac{2 m^2 K_\Theta}{\varepsilon} \log(4)}{n}}$$

$$\leqslant 2 C K_\ell \varepsilon + \overline{M} \sqrt{\frac{2(m+1) \log \left( \frac{16 m R_\Theta}{\varepsilon} \right) + \frac{2 m^2 K_\Theta}{\varepsilon} \log(4)}{n}}.$$

The third step is valid if $\frac{16 m R_\Theta}{\varepsilon} \geqslant 2$. We will shortly take $\varepsilon$ to be equal to $\frac{1}{\sqrt{n}}$, thus this condition holds true under the assumption from the Theorem that $m R_\Theta \sqrt{n} \geqslant 3$. Hence we obtain

$$\mathcal{R}(\widehat{\theta}_n) \leqslant \widehat{\mathcal{R}}_n(\widehat{\theta}_n) + 2 C K_\ell \varepsilon + \overline{M} \sqrt{\frac{2(m+1) \log \left( \frac{16 m R_\Theta}{\varepsilon} \right) + \frac{2 m^2 K_\Theta}{\varepsilon} \log(4)}{n}} + \frac{\overline{M} \sqrt{2}}{\sqrt{n}} \sqrt{\log \frac{1}{\delta}}. \tag{16}$$

Now denote $\tilde{B} = 2 \overline{M} K_f \exp(K_f R_\Theta)$. Then $C K_\ell \leqslant \tilde{B}$ and $2 \overline{M} \leqslant \tilde{B}$. Taking $\varepsilon = \frac{1}{\sqrt{n}}$, we obtain

$$\mathcal{R}(\widehat{\theta}_n) \leqslant \widehat{\mathcal{R}}_n(\widehat{\theta}_n) + \frac{2 \tilde{B}}{\sqrt{n}} + \frac{\tilde{B}}{2} \sqrt{\frac{2(m+1) \log(16 m R_\Theta \sqrt{n})}{n} + \frac{2 m^2 K_\Theta \log(4)}{\sqrt{n}}} + \frac{\tilde{B}}{\sqrt{n}} \sqrt{\log \frac{1}{\delta}}$$

$$\leqslant \widehat{\mathcal{R}}_n(\widehat{\theta}_n) + \frac{2 \tilde{B}}{\sqrt{n}} + \frac{\tilde{B}}{2} \sqrt{\frac{2(m+1) \log(16 m R_\Theta \sqrt{n})}{n}} + \frac{\tilde{B}}{2} \frac{m \sqrt{2 K_\Theta \log(4)}}{n^{1/4}} + \frac{\tilde{B}}{\sqrt{n}} \sqrt{\log \frac{1}{\delta}}$$

$$\leqslant \widehat{\mathcal{R}}_n(\widehat{\theta}_n) + \frac{3 \tilde{B}}{2} \sqrt{\frac{2(m+1) \log(16 m R_\Theta \sqrt{n})}{n}} + \tilde{B} \frac{m \sqrt{K_\Theta}}{n^{1/4}} + \frac{\tilde{B}}{\sqrt{n}} \sqrt{\log \frac{1}{\delta}},$$

since $2 \leqslant 2\sqrt{\log(2)} \leqslant \sqrt{2(m+1) \log(16 m R_\Theta \sqrt{n})}$ since $16 m R_\Theta \sqrt{n} \geqslant 2$ by the Theorem's assumptions, and $\sqrt{2 \log(4)} \leqslant 2$. We finally obtain that

$$\mathcal{R}(\widehat{\theta}_n) \leqslant \widehat{\mathcal{R}}_n(\widehat{\theta}_n) + 3 \tilde{B} \sqrt{\frac{(m+1) \log(m R_\Theta n)}{n}} + \tilde{B} \frac{m \sqrt{K_\Theta}}{n^{1/4}} + \frac{\tilde{B}}{\sqrt{n}} \sqrt{\log \frac{1}{\delta}},$$

by noting that $n \geqslant 9 \max(m^{-2} R_{\Theta}^{-2}, 1)$ implies that

$$\log(16mR_{\Theta}\sqrt{n}) \leqslant 2\log(mR_{\Theta}n).$$

The result unfolds since the constant $B$ in the Theorem is equal to $3\tilde{B}$.

## A.5 Proof of Corollary 1

The corollary is an immediate consequence of Theorem 1. To obtain the result, note that $m = d^2$, thus in particular $\sqrt{m+1} = \sqrt{d^2 + 1} \leqslant d + 1$, and besides $\log(R_{\mathcal{W}}d^2 n) \leqslant 2\log(R_{\mathcal{W}}dn)$ since $R_{\mathcal{W}}n \leqslant R_{\mathcal{W}}^2 n^2$ by assumption on $n$.

## A.6 Proof of Proposition 4

For $x \in \mathcal{X}$, let $(H_k)_{0 \leqslant k \leqslant L}$ be the values of the layers defined by the recurrence (10) with the weights $\mathbf{W}$ and the input $H_0 = x$. We denote by $\|\cdot\|$ the $\ell_2$-norm for vectors and the spectral norm for matrices. Then, for $k \in \{0, \ldots, L-1\}$, we have

$$\|H_{k+1}\| \leqslant \|H_k\| + \frac{1}{L}\|W_k\sigma(H_k)\| \leqslant \|H_k\| + \frac{1}{L}\|W_k\|\,\|\sigma(H_k)\| \leqslant \left(1 + \frac{K_\sigma R_{\mathcal{W}}}{L}\right)\|H_k\|,$$

where the last inequality uses that the spectral norm of a matrix is upper-bounded by its $(1,1)$-norm and that $\sigma(0) = 0$. As a consequence, for any $k \in \{0, \ldots, L\}$,

$$\|H_k\| \leqslant \left(1 + \frac{K_\sigma R_{\mathcal{W}}}{L}\right)^k \|H_0\| \leqslant \exp(K_\sigma R_{\mathcal{W}})R_{\mathcal{X}} =: C,$$

yielding the first claim of the Proposition.

Now, let $\tilde{H}$ be the values of the layers (10) with another parameter $\tilde{\mathbf{W}}$ and with the same input $\tilde{H}_0 = x$. Then, for any $k \in \{0, \ldots, L-1\}$,

$$H_{k+1} - \tilde{H}_{k+1} = H_k - \tilde{H}_k + \frac{1}{L}(W_k\sigma(H_k) - \tilde{W}_k\sigma(\tilde{H}_k)).$$

Hence, using again that the spectral norm of a matrix is upper-bounded by its $(1,1)$-norm and that $\sigma(0) = 0$,

$$\|H_{k+1} - \tilde{H}_{k+1}\| \leqslant \|H_k - \tilde{H}_k\| + \frac{1}{L}\|W_k(\sigma(H_k) - \sigma(\tilde{H}_k))\| + \frac{1}{L}\|(W_k - \tilde{W}_k)\sigma(\tilde{H}_k)\|$$

$$\leqslant \left(1 + K_\sigma \frac{R_{\mathcal{W}}}{L}\right)\|H_k - \tilde{H}_k\| + \frac{K_\sigma}{L}\|W_k - \tilde{W}_k\|\,\|\tilde{H}_k\|$$

$$\leqslant \left(1 + K_\sigma \frac{R_{\mathcal{W}}}{L}\right)\|H_k - \tilde{H}_k\| + \frac{CK_\sigma}{L}\|W_k - \tilde{W}_k\|.$$

Then, dividing by $(1 + K_\sigma \frac{R_{\mathcal{W}}}{L})^{k+1}$ and using the method of differences, we obtain that

$$\frac{\|H_k - \tilde{H}_k\|}{(1 + K_\sigma \frac{R_{\mathcal{W}}}{L})^k} \leqslant \|H_0 - \tilde{H}_0\| + \frac{CK_\sigma}{L}\sum_{j=0}^{k-1}\frac{\|W_j - \tilde{W}_j\|}{(1 + K_\sigma \frac{R_{\mathcal{W}}}{L})^{j+1}}$$

$$\leqslant \frac{CK_\sigma}{L}\|\mathbf{W} - \tilde{\mathbf{W}}\|_{1,1,\infty}\sum_{j=0}^{k-1}\frac{1}{(1 + K_\sigma \frac{R_{\mathcal{W}}}{L})^{j+1}}.$$

Finally note that

$$\sum_{j=0}^{k-1}\frac{(1 + K_\sigma \frac{R_{\mathcal{W}}}{L})^k}{(1 + K_\sigma \frac{R_{\mathcal{W}}}{L})^{j+1}} = \sum_{j=0}^{k-1}(1 + K_\sigma \frac{R_{\mathcal{W}}}{L})^j$$

$$= \frac{L}{K_\sigma R_{\mathcal{W}}}\left((1 + K_\sigma \frac{R_{\mathcal{W}}}{L})^k - 1\right)$$

$$\leqslant \frac{L}{K_\sigma R_{\mathcal{W}}}(\exp(K_\sigma R_{\mathcal{W}}) - 1).$$

We conclude that

$$\|H_k - \tilde{H}_k\| \leqslant \frac{C}{R_{\mathcal{W}}}(\exp(K_\sigma R_{\mathcal{W}}) - 1)\|\mathbf{W} - \tilde{\mathbf{W}}\|_{1,1,\infty} \leqslant \frac{R_{\mathcal{X}}}{R_{\mathcal{W}}}\exp(2K_\sigma R_{\mathcal{W}})\|\mathbf{W} - \tilde{\mathbf{W}}\|_{1,1,\infty}.$$

## A.7 Proof of Proposition 5

For two integers $a$ and $b$, denote respectively $a//b$ and $a\%b$ the quotient and the remainder of the Euclidean division of $a$ by $b$. Then, for $\mathbf{W} \in \mathbb{R}^{L \times d \times d}$, let $\phi(\mathbf{W}) : [0,1] \to \mathbb{R}^{d^2}$ the piecewise-affine function defined as follows: $\phi(\mathbf{W})$ is affine on every interval $\left[\frac{k}{L}, \frac{k+1}{L}\right]$ for $k \in \{0, \ldots, L-1\}$; for $k \in \{1, \ldots, L\}$ and $i \in \{1, \ldots, d^2\}$,

$$\phi(\mathbf{W})_i\left(\frac{k}{L}\right) = \mathbf{W}_{\frac{k}{L},(i//d)+1,(i\%d)+1},$$

and $\phi(\mathbf{W})_i(0) = \phi(\mathbf{W})_i(1/L)$. Then $\phi(\mathbf{W})$ satisfies two properties. First, it is a linear function of $\mathbf{W}$. Second, for $\mathbf{W} \in \mathbb{R}^{L \times d \times d}$,

$$\|\phi(\mathbf{W})\|_{1,\infty} = \|\mathbf{W}\|_{1,1,\infty},$$

because, for $x \in [0,1]$, $\phi(\mathbf{W})(x)$ is a convex combination of two vectors that are bounded in $\ell_1$-norm by $\|\mathbf{W}\|_{1,1,\infty}$, so it is itself bounded in $\ell_1$-norm by $\|\mathbf{W}\|_{1,1,\infty}$, implying that $\|\phi(\mathbf{W})\|_{1,\infty} \leqslant \|\mathbf{W}\|_{1,1,\infty}$. Reciprocally,

$$\|\phi(\mathbf{W})\|_{1,\infty} = \sup_{0 \leqslant t \leqslant 1} \|\phi(\mathbf{W})(x)\|_1 \geqslant \sup_{1 \leqslant k \leqslant L} \left\|\phi(\mathbf{W})\left(\frac{k}{L}\right)\right\|_1 = \|\mathbf{W}\|_{1,1,\infty}.$$

Now, take $\mathbf{W} \in \mathcal{W}$. The second property of $\phi$ implies that $\|\phi(\mathbf{W})\|_{1,\infty} \leqslant R_{\mathcal{W}}$. Moreover, each coordinate of $\phi(\mathbf{W})$ is $K_{\mathcal{W}}$-Lipschitz, since the slope of each piece of $\phi(\mathbf{W})_i$ is at most $K_{\mathcal{W}}$. As a consequence, $\phi(\mathbf{W})$ belongs to

$$\Theta_{\mathcal{W}} = \{\theta : [0,1] \to \mathbb{R}^{d^2}, \|\theta\|_{1,\infty} \leqslant R_{\mathcal{W}} \text{ and } \theta_i \text{ is } K_{\mathcal{W}}\text{-Lipschitz for } i \in \{1, \ldots, d^2\}\}.$$

Therefore $\phi(\mathcal{W})$ is a subset of $\Theta_{\mathcal{W}}$, thus its covering number is less than the one of $\Theta_{\mathcal{W}}$. Moreover, $\phi$ is clearly injective, thus we can define $\phi^{-1}$ on its image. Consider an $\varepsilon$-cover $(\theta_1, \ldots, \theta_N)$ of $(\phi(\mathcal{W}), \|\cdot\|_{1,\infty})$. Let us show that $(\phi^{-1}(\theta_1), \ldots, \phi^{-1}(\theta_N))$ is an $\varepsilon$-cover of $(\mathcal{W}, \|\cdot\|_{1,1,\infty})$: take $\mathbf{W} \in \mathcal{W}$ and consider $\theta_i$ a cover member at distance less than $\varepsilon$ from $\phi(\mathbf{W})$. Then

$$\|\mathbf{W} - \phi^{-1}(\theta_i)\|_{1,1,\infty} = \|\phi(\mathbf{W} - \phi^{-1}(\theta_i))\|_{1,\infty} = \|\phi(\mathbf{W}) - \theta_i\|_{1,\infty} \leqslant \varepsilon,$$

where the second equality holds by linearity of $\phi$. Therefore, the covering number of $(\mathcal{W}, \|\cdot\|_{1,1,\infty})$ is upper bounded by the one of $(\phi(\mathcal{W}), \|\cdot\|_{1,\infty})$, which itself is upper bounded by the one of $(\Theta_{\mathcal{W}}, \|\cdot\|_{1,\infty})$, yielding the result by Proposition 3.

## A.8 Proof of Theorem 2

The proof structure is the same as the one of Theorem 1, but some constants change. Similarly to (16), we obtain that, if $\frac{16d^2 R_{\mathcal{W}}}{\varepsilon} \geqslant 2$ (which holds true for $\varepsilon = 1/\sqrt{n}$ and under the assumption of the Theorem),

$$\mathcal{R}(\widehat{\mathbf{W}}_n) \leqslant \widehat{\mathcal{R}}_n(\widehat{\mathbf{W}}_n) + 2CK_\ell\varepsilon + \overline{M}\sqrt{\frac{2(d^2+1)\log\left(\frac{16d^2 R_{\mathcal{W}}}{\varepsilon}\right) + \frac{2d^4 K_{\mathcal{W}}}{\varepsilon}\log(4)}{n}} + \frac{\overline{M}\sqrt{2}}{\sqrt{n}}\sqrt{\log\frac{1}{\delta}},$$

with

$$\overline{M} = K_\ell(R_{\mathcal{X}}\exp(K_\sigma R_{\mathcal{W}}) + R_{\mathcal{Y}})$$

and

$$C = \frac{R_{\mathcal{X}}}{R_{\mathcal{W}}}\exp(2K_\sigma R_{\mathcal{W}}).$$

Finally denote

$$\tilde{B} = 2\overline{M}\max\left(\frac{\exp(K_\sigma R_{\mathcal{W}})}{R_{\mathcal{W}}}, 1\right).$$

Then $CK_\ell \leqslant \tilde{B}$ and $2\overline{M} \leqslant \tilde{B}$. Taking $\varepsilon = \frac{1}{\sqrt{n}}$, we obtain as in the proof of Theorem 1 that

$$\mathcal{R}(\widehat{\mathbf{W}}_n) \leqslant \widehat{\mathcal{R}}_n(\widehat{\mathbf{W}}_n) + 3\tilde{B}\sqrt{\frac{(d^2+1)\log(d^2 R_{\mathcal{W}} n)}{n}} + \tilde{B}\frac{d^2\sqrt{K_{\mathcal{W}}}}{n^{1/4}} + \frac{\tilde{B}}{\sqrt{n}}\sqrt{\log\frac{1}{\delta}}.$$

for $n \geqslant 9R_{\mathcal{W}}^{-1} \max(d^{-4}R_{\mathcal{W}}^{-1}, 1)$. Thus

$$\mathscr{R}(\widehat{\mathbf{W}}_n) \leqslant \widehat{\mathscr{R}}_n(\widehat{\mathbf{W}}_n) + 3\sqrt{2}\tilde{B}(d+1)\sqrt{\frac{\log(dR_{\mathcal{W}}n)}{n}} + \tilde{B}\frac{d^2\sqrt{K_{\mathcal{W}}}}{n^{1/4}} + \frac{\tilde{B}}{\sqrt{n}}\sqrt{\log\frac{1}{\delta}},$$

since $\sqrt{d^2+1} \leqslant d+1$ and $R_{\mathcal{W}}n \leqslant R_{\mathcal{W}}^2 n^2$ by assumption on $n$. The result unfolds since the constant $B$ in the Theorem is equal to $3\sqrt{2}\tilde{B}$.

### A.9 Proof of Corollary 2

Let

$$A(\mathbf{W}) = \left(\prod_{k=1}^{L}\left\|I + \frac{1}{L}W_k\right\|\right)\left(\sum_{k=1}^{L}\frac{\|W_k^T\|_{2,1}^{2/3}}{L^{2/3}\|I + \frac{1}{L}W_k\|^{2/3}}\right)^{3/2},$$

where $\|\cdot\|_{2,1}$ denotes the $(2,1)$-norm defined as the $\ell_1$-norm of the $\ell_2$-norms of the columns, and $I$ is the identity matrix (and we recall that $\|\cdot\|$ denotes the spectral norm). We apply Theorem 1.1 from Bartlett et al. (2017) by taking as reference matrices the identity matrix. The theorem shows that, under the assumptions of the corollary,

$$\mathbb{P}\left(\underset{1\leqslant j\leqslant d}{\arg\max}\, F_{\mathbf{W}}(x)_j \neq y\right) \leqslant \widehat{\mathscr{R}}_n(\mathbf{W}) + C\frac{R_{\mathcal{X}}A(\mathbf{W})\log(d)}{\gamma\sqrt{n}} + \frac{C}{\sqrt{n}}\sqrt{\log\frac{1}{\delta}},$$

where, as in the corollary, $\widehat{\mathscr{R}}_n(\mathbf{W}) \leqslant n^{-1}\sum_{i=1}^{n}\mathbf{1}_{F_{\mathbf{W}}(x_i)_{y_i}\leqslant\gamma+\max_{j\neq y_i}f(x_i)_j}$ and $C$ is a universal constant. Let us upper bound $A(\mathbf{W})$ to conclude. On the one hand, we have

$$\prod_{k=1}^{L}\left\|I + \frac{1}{L}W_k\right\| \leqslant \prod_{k=1}^{L}\left(\|I\| + \frac{1}{L}\|W_k\|\right)$$

$$\leqslant \prod_{k=1}^{L}\left(1 + \frac{1}{L}\|W_k\|_{1,1}\right)$$

$$\leqslant \prod_{k=1}^{L}\left(1 + \frac{1}{L}R_{\mathcal{W}}\right)$$

$$\leqslant \exp(R_{\mathcal{W}})$$

On the other hand, for any $k \in \{1,\ldots,L\}$,

$$\|W_k^T\|_{2,1} \leqslant \|W_k^T\|_{1,1} \leqslant R_{\mathcal{W}},$$

while

$$\left\|I + \frac{1}{L}W_k\right\| \geqslant 1 - \frac{1}{L}\|W_k\| \geqslant 1 - \frac{R_{\mathcal{W}}}{L} \geqslant \frac{1}{2},$$

under the assumption that $L \geqslant R_{\mathcal{W}}$. All in all, we obtain that

$$A(\mathbf{W}) \leqslant \exp(R_{\mathcal{W}})\left(2^{2/3}L^{1/3}R_{\mathcal{W}}^{2/3}\right)^{3/2} = 2R_{\mathcal{W}}\exp(R_{\mathcal{W}})\sqrt{L},$$

which yields the result.

## B   Experimental details

Our code is available at

We use the following model, corresponding to model (10) with additional projections at the beginning and at the end:

$$H_0 = Ax$$

$$H_{k+1} = H_k + \frac{1}{L}W_{k+1}\sigma(H_k), \quad 0 \leqslant k \leqslant L-1$$

$$F_{\mathbf{W}}(x) = BH_L,$$

| Name | Value |
|:---:|:---:|
| $d$ | 30 |
| $L$ | 1000 |
| $\sigma$ | ReLU |

Table 1: Values of the model hyperparameters.

where $x \in \mathbb{R}^{768}$ is a vectorized MNIST image, $A \in \mathbb{R}^{d \times 768}$, and $B \in \mathbb{R}^{10 \times d}$. Table 1 gives the value of the hyperparameters.

We use the initialization scheme outlined in Section 4.1: we initialize, for $k \in \{1, \ldots, L\}$ and $i, j \in \{1, \ldots, d\}$,

$$\mathbf{W}_{k,i,j} = \frac{1}{\sqrt{d}} f_{i,j}\left(\frac{k}{L}\right),$$

where $f_{i,j}$ are independent Gaussian processes with the RBF kernel (with bandwidth equal to 0.1). We refer to Marion et al. (2022) and Sander et al. (2022) for further discussion on this initialization scheme. However, $A$ and $B$ are initialized with a more usual scheme, namely with i.i.d. $\mathcal{N}(0, 1/c)$ random variables, where $c$ denotes the number of columns of $A$ (resp. $B$).

In Figure 1a, we repeat training 10 times independently. Each time, we perform 30 epochs, and compute after each epoch both the Lipschitz constant of the weights and the generalization gap. This gives 300 pairs (Lipschitz constant, generalization gap), which each corresponds to one dot in the figure. Furthermore, we report results for two setups: when $A$ and $B$ are trained or when they are fixed random matrices.

In Figure 1b, $A$ and $B$ are not trained. The reason is to assess the effect of the penalization on $\mathbf{W}$ for a fixed scale of $A$ and $B$. If we allow $A$ and $B$ to vary, then it is possible that the effect of the penalization might be neutralized by a scale increase of $A$ and $B$ during training.

For all experiments, we use the standard MNIST datasplit (60k training samples and 10k testing samples). We train using the cross entropy loss, mini-batches of size 128, and the optimizer Adam (Kingma and Ba, 2015) with default parameters and a learning rate of 0.02.

We use PyTorch (Paszke et al., 2019) and PyTorch Lightning for our experiments.

The code takes about 60 hours to run on a standard laptop (no GPU).

