# OpenReview forum: "Generalization bounds for neural ordinary differential equations and deep residual networks"
_NeurIPS.cc/2023/Conference — NeurIPS 2023 poster_

### Official Review · Reviewer_wvQq · 2023-07-06

**Soundness:** 3 good
**Presentation:** 3 good
**Contribution:** 3 good
**Rating:** 6
**Confidence:** 3

**Summary:**

【Post-rebuttal Comments】

 I thank the authors for the discussions after the authors' rebuttal. My questions are appropriately answered. So, I want to keep my score and vote for acceptance.


【Original Comments】

This paper evaluates the generalization performance of the class of functions represented as solutions of time-dependent parametrized ODEs (including time-dependent Neural ODEs) and their discretized variants, that is, Residual Network, that smoothly changes weight parameters with respect to layer index. The obtained generalization bound is $O(1/n^4)$ for the general case and $O(1/n^2)$ for the time-independent case (for sample size $n$.) Numerical experiments were conducted to examine the relationship between the smoothness of weights between successive layers and generalization performance. Also, a learning method was proposed to add the difference in weights between successive layers as a regularization.

**Strengths:**

- The derived results are consistent with classical statistical learning theory. Specifically, setting $K_\Theta=0$ yields the classical $O(1/n^2)$ rate.
- Numerical experiments verify the relationship between the smoothness of weights and generalization performance. These results align with the theory.
- Comparisons with existing results (Bartlett et al. (2017), Golowich et al. (2018)) are carefully considered. In particular, this study is novel in that it provides generalization bounds for a class of depth-independent models to which Golowich et al. (2018) are not applicable.
- The paper is well-written. The organization and mathematical description of the paper are appropriate, and it was easy to understand the paper's main point.

**Weaknesses:**

- To the best of my knowledge, it is rare to impose smoothness between the weights of successive layers in a discretized ResNet model. In addition, the prediction accuracy of those models has yet to be confirmed to be sufficient. Therefore, it is difficult to say that this paper provides a theoretical guarantee for practical models.

**Questions:**

In l.84--91, this paper refers to existing studies of the generalization bound of continuous-time NNs. However, their relationship with this study has yet to be discussed extensively. Is the novelty of this compared with them that the former is a time-independent ODE while the latter is a time-dependent one? Also, is there any study that dealt with time-independent ODEs?

**Limitations:**

Limitations are discussed in the experiment section (l.338) and conclusion section (l.346.)

---

> ### Author Rebuttal · Authors · 2023-08-08
>
> > To the best of my knowledge, it is rare to impose smoothness between the weights of successive layers in a discretized ResNet model. In addition, the prediction accuracy of those models has yet to be confirmed to be sufficient.
>
> We agree that imposing smoothness between successive layers is not a very common setup in residual networks, although it is the right framework for residual networks to approximate neural ODEs, which are commonly used models. We leave for future work the task of evaluating the prediction accuracy on real-world data of residual networks in this setting.
>
> > In l.84--91, this paper refers to existing studies of the generalization bound of continuous-time NNs. However, their relationship with this study has yet to be discussed extensively. Is the novelty of this compared with them that the former is a time-independent ODE while the latter is a time-dependent one?
>
> Two of the three papers cited in lines 84--91 tackle recurrent neural networks, which handle time series, contrarily to our residual networks that handles vectors. The time series setup requires significantly different models and analysis. The last paper answers a separate (although related) question regarding the generalization across environments, and only gives a generalization bound for linear ODEs. We will make the comparison with these works more detailed.
>
> > Also, is there any study that dealt with time-independent ODEs?
>
> We were made aware after submission time of another work (reference below) that shows a generalization bound for parametrized ODEs for manifold learning, which applies in particular for neural ODEs. Their proof technique bears some similarities with ours, but the model (time-independent versus time-dependent ODEs), task (unsupervised manifold learning versus supervised learning), as well as the absence of connection with residual networks, differ from our approach. We will of course include the reference and discuss differences in the next version of the paper.
>
> J. Hanson and M. Raginsky. Fitting an immersed submanifold to data via sussmann’s orbit theorem. In 2022 IEEE 61st Conference on Decision and Control (CDC), pages 5323–5328, 2022
>
> We are not aware of any other generalization bound for neural ODEs, be it in the time-dependent or time-independent setting.

---

> > ### Comment · Reviewer_wvQq · 2023-08-15
> >
> > I thank the authors for considering my comments and responding to them.
> >
> > **Smoothness between successive layers**
> > I agree with the authors that while the smoothness assumption between succcessive layers is uncommon, it is natural when we interpret ResNet as a discretization of Neural ODE. I also agree that how the smoothness assumption incurs the prediction performances. Since generalization is not an issue, at least theoretically, the problem may lie in expressive power or optimization.
> >
> >
> > **Relation to prior work on continuous-time NNs**
> > OK
> >
> >
> > **Relation to prior work on time-independent ODEs**
> > OK

---

> > > ### Comment · Reviewer_wvQq · 2023-08-15
> > > **Typos in my initial review comment**
> > >
> > > $O(1/n^4)$ and $O(1/n^2)$ in [my initial review comment](https://openreview.net/forum?id=992vogTP1L&noteId=ALqCd56a4W) should have been $O(n^{-1/4})$ and $O(n^{-1/2})$, respectively. I am sorry if I confused you.

---

> > > > ### Author Response · Authors · 2023-08-15
> > > >
> > > > Thank you again for your review, additional remarks, and for pointing out the typos. We fully agree that the expressive power and optimization of smooth residual networks is an important question, which we leave for future work. We will make sure to mention this in the next version of the paper.

---

### Official Review · Reviewer_EEyp · 2023-07-07

**Soundness:** 4 excellent
**Presentation:** 3 good
**Contribution:** 4 excellent
**Rating:** 7
**Confidence:** 3

**Summary:**

The authors present a generalization bound for a large class of ODEs in this work. They connect ODEs to residual architectures to control the generalization with the differences between weight matrices.

**Strengths:**

The authors present a theoretical bound, which is something worth highlighting when everybody is just building ad hoc pipelines and then testing them against some benchmark. From this point of view, this work represents a clear advance over other results.

**Weaknesses:**

Even though the results are strong, the consequences and how they can be used are not done in a way that can help the reader to fully understand the applications of these results.

**Questions:**

Have you measured how the bounds fare in time series problems?

**Limitations:**

None.

---

> ### Author Rebuttal · Authors · 2023-08-08
>
> We consider in this paper a residual network model that is not adapted for time series: the input is a vector $x \in \mathbb{R}^d$ that is used on the first layer of the network. On the contrary, models for time series typically input data at each layer, corresponding to a new time step. The time series setup requires different models (such as ODE-like RNNs, which are mentioned in the related work section), which are not in the scope of the current paper. We will specify this more clearly in the related work section where we mention RNNs, to avoid inducing any confusion.

---

### Official Review · Reviewer_d5dy · 2023-07-07

**Soundness:** 4 excellent
**Presentation:** 3 good
**Contribution:** 3 good
**Rating:** 7
**Confidence:** 4

**Summary:**

The paper explores the generalization ability of neural ordinary differential equations and deep residual networks through Lipschitz-based complexity arguments. The bound, specifically for the discretized version, involves the maximum magnitude of the difference between successive weight matrices, which is not commonly seen. The paper also uses numerical experiments to investigate how this quantity relates to the generalization capability of deep residual networks. The generalization gap is smaller when the quantity is smaller, but adding the quantity as a penalization term did not improve the prediction performance.

**Strengths:**

[originality]

- To the best of my knowledge, this paper is the first to present a generalization error bound for neural ODEs. The proof techniques used in this work are rather standard, but they are combined in new ways to derive the results.
- The potential importance of the Lipschitzness of the weights (i.e., the maximal magnitude of the differences between successive weight matrices) is newly proposed.

[quality]

- The analysis and results seem reasonable, and the mathematical setup is adequately explained.

[clarity]

- The paper is well-organized and articulately written. Its concise abstract enhances the paper's accessibility and ease of comprehension.

[significance]

- Neural ODEs and deep residual networks are now among the standard tools in the machine learning community. Therefore, theoretical results that advance our understanding of the behavior of these models are relevant and can improve their reliability for use in various applications.
- The analysis in Theorem 1 reveals a term whose convergence rate is O(n^{-1/4}). Corollary 1 exemplifies that the slow-rate term can be eliminated by making the coefficient parameter time-independent in the infinite-dimensional case, partly elucidating the consequences of having time-dependent component in the model.

**Weaknesses:**

[originality]

- None in particular

[quality]

- The paper would benefit from additional citations, particularly where named theorems are mentioned in the main text. For instance, it would be helpful to add citations to the Picard-Lindelof theorem at Line 59 and to Gronwall's inequality at Line 166 for the convenience of the reader.

[clarity]

- None in particular

[significance]

- None in particular

**Questions:**

Have you conducted any experiments of Figure 1b using the maximum max-norm of the weights as a penalty term (or a sum of the max-norm of the weights) instead of the squared sum of Frobenius norms? While I acknowledge the equivalence between matrix norms, and although it may not be strictly necessary for the paper, I believe that readers (including myself) would be interested in seeing whether the behavior is different when the quantity that appeared directly in deriving the theoretical guarantee is used.

[minor suggestions]

- Line 112 “takes values into” → “takes values in”
- Line 285 “Corollary 2 (of Theorem 1.1 of Bartlett et al. (2017))” → “Corollary 2 (corollary of Theorem 1.1 of Bartlett et al. (2017))”
- Line 319 “a.k.a.” → “i.e.,” may be more appropriate?

**Limitations:**

None in particular

---

> ### Author Rebuttal · Authors · 2023-08-08
>
> > The paper would benefit from additional citations, particularly where named theorems are mentioned in the main text.
>
> We will add citations for the Picard-Lindelöf theorem (actually the citation is present in the appendix but not in the main paper) and Grönwall’s inequality.
>
> > Have you conducted any experiments of Figure 1b using the maximum max-norm of the weights as a penalty term (or a sum of the max-norm of the weights) instead of the squared sum of Frobenius norms?
>
> We had not previously conducted an experiment with the max-norm penalization, because we thought that this penalization may be too irregular, in the sense that, at any one step of the backpropagation, it only impacts the maximum weights and not the others. Nevertheless, this is a very relevant remark, and we performed the suggested experiment. The results are mixed: we observe a similar effect of the regularization on the generalization gap as in the paper when using the $L_2$ norm of the max-norm of the weights. On the other hand, penalizing by the maximum max-norm of the weights does not have an effect on the generalization gap. We interpret this last result as a consequence of the fact that the maximum max-norm is too irregular (it only acts on two scalar weights at each step) to be used in practice. We will report these results and discussion in the next version of the paper.
>
> The minor suggestions are well noted.

---

> > ### Comment · Reviewer_d5dy · 2023-08-11
> > **Thank you for the response**
> >
> > I have read the authors' responses, and points have been noted. I have no further questions, and my evaluation of the paper did not change.

---

### Official Review · Reviewer_W2zB · 2023-07-09

**Soundness:** 4 excellent
**Presentation:** 4 excellent
**Contribution:** 4 excellent
**Rating:** 6
**Confidence:** 4

**Summary:**

The paper provides a generalization bound for the large class of time-dependent and time-independent neural ODEs. In addition, by leveraging on the connection between neural ODEs and deep residual networks, the paper provides a depth-independent generalization bound for the class of deep residual networks. The bound is compared with some earlier results, showing its novelty. The paper introduces a novel way of controlling the statistical complexity of neural networks and the bound depends on the magnitude of the difference between successive weight matrices. Numerical experiments are provided to show the relationship between this bound and the generalization ability of neural networks.

**Strengths:**

The paper provides a generalization bound for the large class of time-dependent, time-independent neural ODEs and deep residual networks. The bound is compared with some earlier results, showing its novelty. The paper introduces a novel way of controlling the statistical complexity of neural networks and the bound depends on the magnitude of the difference between successive weight matrices. Numerical experiments are provided to show the relationship between this bound and the generalization ability of neural networks.

**Weaknesses:**

Some expressions are not very clear, and some results have too strong conditions. Please see Questions below.

**Questions:**

1. In Proposition 2 and Theorem 1, what do $R_X$ and $R_Y$ mean? The reviewer guesses that $R_X$ is the bound of the set of initial values $x$, but these two notations as well as $X$ and $Y$ do not have clear definitions.

2. A clear definition of $n$ in Theorem 1 is missing. If the data is any $n$ samples, the conclusion of Theorem 1 is obviously wrong.

3. The condition in Corollary 1 is too strong. The reviewer think Neural ODE should contains at least one hidden layer. And the reviewer think that the generalization of Neural ODE with one hidden layer can be obtained by using Theorem 1, by studying the Lipschitz constant of $\sigma(Wx)$.

4. The reviewer thinks that some existing definitions or symbols, such as covering number, subGaussian, should be briefly explained.

5. The numerical experiments show the correlation between the generalization gap and the maximum Lipschitz constant of the weight. However, the generalization gap proved in this paper mainly depends on the number of samples $n$. Could the paper provide experiments to show the correlation between these two quantities as well as the convergence order?

6. Some abnormal result points can be found in the experimental results, such as the point in the bottom left corner of Figure 1. Is there any explanation for this？

7. Theory and numerical experiments show small Lipschitz constant of the weights reduce the generalization gap. Does this indicate that time independent coefficients ($K_{\Theta}=0$?) have better generalization?



**Limitations:**

The authors have adequately addressed the limitations.

---

> ### Author Rebuttal · Authors · 2023-08-08
>
> 1. The notations $X$, $Y$, $R_X$ and $R_Y$ are defined in Section 3.1 (lines 111-113).
> 2. $n$ is indeed the sample size (defined in Section 3.1, line 111), and the training sample is drawn i.i.d. from the same distribution as the test sample (as specified in Section 3.1). Does this answer your concern on Theorem 1? If not, we would be very interested if you could elaborate on your statement.
> 3. Our parameterized ODE model (5) does not include the case where there are weights inside the non-linearity, since we assume the dynamics at time $t$ to be linear with respect to the parameters (which still makes the input-output mapping highly nonlinear, see Section 3.2, lines 150-154). As a consequence, the extension to the case you mention is non trivial, and we agree it would be very interesting. We leave it for future work and will mention this possible extension in the conclusion of the paper.
> 4. The covering number is informally defined on lines 172-174, we will add a more formal definition, as well as the definition of sub-Gaussianity.
> 5. Following your suggestion, we performed some experiments varying the sample size $n$. We observe a smaller generalization gap when increasing $n$, as expected by the theory. Unfortunately, it is difficult to say more and in particular to report a convergence rate because of a large amount of noise in the experiments (due to the data splitting used to vary the sample size and to the optimization algorithm).
> 6. The points in the bottom left corner of Figure 1a correspond to a very small number of training epochs (typically equal to 1). At this early stage in the training process, the model is underfitted, and the generalization gap is very negative.
> 7. Your statement is correct: we do observe that time independent coefficients have better generalization (however at the cost of a less expressive model). We will report this in the experiments by adding data points corresponding to time-independent coefficients (which can be thought of as taking $\lambda$ to infinity in Figure 1b).

---

> > ### Comment · Reviewer_W2zB · 2023-08-14
> >
> > According to the author's response and my reading of the proof, my question has been effectively addressed. However, I would like to suggest adding rigorous definition of the symbols (e.g., $n$) in the theorems.

---

> > > ### Author Response · Authors · 2023-08-15
> > >
> > > Thank you again for your review and additional suggestion. In the next version, we will recall the definitions before the theorems or refer to Section 3.1 to remove any ambiguity.

---

### Author Rebuttal · Authors · 2023-08-08

Dear reviewers,

We warmly thank you for your time and relevant comments, which will help us improve our work. If accepted, we intend to take into account your suggestions, making use of the additional page.

We answer the specifics of questions pointed out by the reviewers in individual responses.

Thank you again,

Sincerely,

The authors

---

### Decision · Program_Chairs · 2023-09-21

**Decision:**

Accept (poster)

**Comment:**

This paper provides generalization bounds for neural ODEs and ResNets. The reviewers appreciate the novel and solid technical contributions, and unanimously recommend acceptance and weak acceptance. The questions from the initial reviews were satisfactorily addressed.